# Rapid Fabrication of Wavelength-Scale Micropores on Metal by Femtosecond MHz Burst Bessel Beam Ablation

**DOI:** 10.3390/nano12244378

**Published:** 2022-12-08

**Authors:** Yang Cheng, Yu Lu, Qing Yang, Jun Zhong, Mengchen Xu, Xiaodan Gou, Lin Kai, Xun Hou, Feng Chen

**Affiliations:** 1School of Mechanical Engineering, Xi’an Jiaotong University, Xi’an 710049, China; 2State Key Laboratory for Manufacturing System Engineering, Shaanxi Key Laboratory of Photonics Technology for Information, School of Electronic Science and Engineering, Xi’an Jiaotong University, Xi’an 710049, China

**Keywords:** femtosecond laser, MHz burst Bessel beam, porous metal, wavelength-scale micropores, laser-induced periodic surface structure (LIPSS)

## Abstract

The preparation of the wavelength-scale micropores on metallic surfaces is limited by the high opacity of metal. At present, most micropores reported in the literature are more than 20 µm in diameter, which is not only large in size, but renders them inefficient for processing so that it is difficult to meet the needs of some special fields, such as aerospace, biotechnology, and so on. In this paper, the rapid laser fabrications of the wavelength-scale micropores on various metallic surfaces are achieved through femtosecond MHz burst Bessel beam ablation. Taking advantage of the long-depth focal field of the Bessel beam, high-density micropores with a diameter of 1.3 µm and a depth of 10.5 µm are prepared on metal by MHz burst accumulation; in addition, the rapid fabrication of 2000 micropores can be achieved in 1 s. The guidelines and experimental results illustrate that the formations of the wavelength-scale porous structures are the result of the co-action of the laser-induced periodic surface structure (LIPSS) effect and Bessel beam interference. Porous metal can be used to store lubricant and form a lubricating layer on the metallic surface, thus endowing the metal resistance to various liquids’ adhesion. The microporous formation process on metal provides a new physical insight for the rapid preparation of wavelength-scale metallic micropores, and promotes the application of porous metal in the fields of catalysis, gas adsorption, structural templates, and bio-transportation fields.

## 1. Introduction

Metal with microporous structure has been attracting great attention due to its many applications ranging from aerospace and biomedical to electronics [1,2,3,4]. The preparation of precise micropores on metal can endow metals with some special properties; for example, it can improve heat dissipation to enhance the service life of aerospace materials [5,6], to increase the anti-reflection performance of material to be applied in stealth [7,8,9], to reinforce the fixation of the lubricating layer to prepare an anti-adhesion surface [10,11], and to strengthen the adhesion of cells to improve the biocompatibility of materials in the biomedical field [12,13,14]. The existing methods of micropores fabrication mainly include anodizing [15], ion beam etching [16,17], femtosecond laser drilling, and so on [10,11], but these methods have certain limitations, such as poor uniformity, low processing efficiency, and high cost. The ultra-short pulse width and ultra-high peak power of femtosecond laser cause the thermal effect to be approximately negligible during processing [18,19,20,21,22], thus femtosecond laser has become one of the most effective methods for ultra-precision machining at present [23,24,25,26]. For example, Jiang et al. used femtosecond laser double pulse to fabricate porous structures with a diameter of 20 µm on NiTi alloy [26]. The drilling speed was 47 pores per second. Lee et al. used femtosecond multi-pulses to fabricate micropores with diameter of 7 µm on titanium implants to enhance the quality of the peri-implant soft-tissue integration [27]. Every spot was created by 2 × 10^4^ pulses, and a spherical focusing lens with a numerical aperture of 0.39 was optimal for the fabrication of pores. Through Bessel femtosecond laser, Stoain et al. fabricated micro-holes with a diameter of 24 µm on stainless steel [28]. The micro-holes preparation was achieved by repeated stacking of multiple pulses while the metal remained stationary. The fastest preparation for single pore required 2 s. At present, ultrashort pulse laser exhibits unique advantages in the preparation of precision micropores [29,30,31], and most of the micropores’ diameters prepared by femtosecond laser on the metallic surface are wider than 20 µm, and the low processing efficiency renders it difficult to achieve rapid processing of a large area; however, many important fields such as aerospace, biochips, medical, and other fields not only require pore size of wavelength scale, but also need very high-density pore array structures; therefore, studying the microporous formation process on a metallic surface has important guiding significance for the rapid preparation of high-density and micro-porous structures.

A large number of active electrons on the metallic surface accelerate the absorption of laser energy on the surface so that it is hard for the laser to penetrate the metal and to propagate to depth and to direction inside [32]. As a result, it is difficult for conventional femtosecond lasers to achieve in-depth energy deposition on metals, leading to micropores being formed by the superposition of single or a few pulses [33]. In this article, a spatiotemporally modulated femtosecond MHz burst Bessel beam is used for fabricating a microporous structure, through which wavelength-scale high-density micropores are rapidly fabricated on metallic surfaces. Prepared high-density micropores have a dimeter of 1.3 µm and a depth of up to 10.5 µm on the surfaces of NiTi alloy. 2000 micropores can be quickly prepared in 1 s. Similar results are seen on aluminum (Al) and stainless steel (SS). These results not only confirm that the formation of wavelength-scale porous structure is the result of the co-action of the laser-induced periodic surface structure (LIPSS) effect and the Bessel beam interference, but also demonstrate the universality of the method for preparing wavelength-scale micropores on metallic surfaces through femtosecond Bessel beam ablation. The microporous formation process of Bessel beam on metal provides a preliminary theoretical reference for the rapid preparation of wavelength-scale micropores on metal, and promotes the application of porous metal in the fields of catalysis, gas adsorption, structural templates, and bio-transportation electrochemistry.

## 2. Materials and Methods

### 2.1. Porous Fabrication

The processing diagram of porous metal is shown in Figure 1a. A femtosecond fiber laser (FemtoYSL-20, YSL-photonics, Wuhan, China) generates a Gaussian beam with the center wavelength of 1030 nm and the pulse width of 270 fs. Output femtosecond pulses are controlled by the square wave signal generated by an external trigger as shown in Figure 1b. When the external trigger is at the high level of 5 V, the laser is produced. When the external trigger is at the low level of 0 V, the laser stops being produced; therefore, the number of laser pulses contained in a single burst (N) can be regulated by adjusting the duty of the square wave, and the frequency of the output pulse train can be regulated by adjusting the square wave frequency; subsequently, the Gaussian beam is converted into Bessel beam through the axicon (Thorlabs, AX-250, α = 1°, Newton, NJ, USA), and then converged into the objective lens (Nikon, ×20, NA = 0.4, Tokyo, Japan). Finally, the Bessel beam is focused on the metallic surface. The movement of the three-dimensional (3D) translation stage drives the movement of metallic samples to achieve the rapid fabrication of the microporous structure. Taking NiTi alloy (Baoji Seabird Metal Material Co., Ltd., Baoji, China) as an example, when the frequency of the square wave generated by the external trigger is 2 kHz, the number of burst pulses is 250. The moving speed of the translation stage is 12,000 µm/s and the fast processing of micropores with high-filling density can be achieved at the speed of 2000 micropores per second.

### 2.2. Characterization

The surface morphologies and elements distribution are observed by scanning electron microscope (SEM, Flex1000, Hitachi, Tokyo, Japan). The depth of the surface structure is obtained by atomic force microscopy (AFM, FM-Nanoview 1000, Zhengzhou, China). The 3D profile of samples is observed by laser confocal microscopy (LEXT-OLS4000, Olympus, Tokyo, Japan).

### 2.3. Anti-Adhesion Process

Silicone oil (Aladdin, PMX-200, Beijing, China) is injected into the porous metal and then vacuumed. The silicone oil seeps into the porous structure and is trapped tightly in the porous structure by capillary forces, eventually forming a lubricating layer on the surface. The anti-adhesion properties of the lubricant-infused porous metal are tested by the sliding performance of milk, cola, coffee, and tomato juice.

### 2.4. Numerical Simulation

In the simulation of this paper, the propagation of the Bessel beam can be described by a unidirectional envelope propagation function:(1)$$\frac{\partial E}{\partial z}=\frac{i}{2k}\left(\frac{\partial^{2}}{\partial r^{2}}+\frac{1}{r}\frac{\partial }{\partial r}\right)E$$
which is solved by a 2D finite difference method. A 50 μm (*r* direction) × 150 μm (*z* direction) square area is chosen as the simulation area, which is divided by 4999 × 500 grids. A perfect match layer is set around the simulation area. The electric field is focused on the top layer of the simulation area, and can be described as:(2)$$E\left(r,0\right)=J_{0}\left(kr\sin\theta \right)\cdot \varepsilon_{Rc}\left(r\right)$$
where $\varepsilon_{Rc}\left(r\right)=1$ when $r<R_{c}$ and $\varepsilon_{Rc}\left(r\right)=0$ when $r>R_{c}$. The value of $R_{c}$ *Rc* is 1 μm when only the core of the Bessel beam is permitted to pass and 2.5 μm when the core and the first ring of the Bessel beam can pass.

## 3. Results and Discussion

The processing diagram of porous metal is shown in Figure 1a. A fiber laser output comprising femtosecond pulses is controlled by the square wave signal generated by the external trigger. When the external trigger is at the high level of 5 V, the laser is produced at a repetition rate of 2.5 MHz. When the external trigger is at the low level of 0 V, the laser stops being produced; therefore, the repetition frequency of MHz burst mode and the number of burst pulses can be controlled by adjusting the square wave frequency and duty output by external trigger, respectively (Figure 1b). Subsequently, the Gaussian beam is converted into Bessel beam through the axicon, and then converged into the objective lens. Finally, the Bessel beam focuses on the metallic surface and achieves the rapid fabrication of the microporous structure through the movement of the translation stage. Take NiTi alloy as an example, the surface morphologies and elements’ distribution of porous NiTi alloy are shown in Figure 1c. Uniform micropores are densely arranged on the metallic surface after laser ablation. The area of 1 cm^2^ can be filled with 4 × 10^4^ pores. The spacing between the micropores is 6 µm. The magnified SEM displays that micropores are trumpet-shaped with a diameter of 2.8 µm. Some nanoparticles are distributed around every micropore. Elemental analysis is shown in Figure 1d; untreated NiTi alloy is mainly composed of Ni, Ti, C, and O elements. The microporous NiTi alloy after laser ablation is still dominated by these four elements, but the intensity of O increases significantly, indicating that NiTi alloy was oxidized after laser ablation (Figure 1e) [34]. 

Bessel beam ablation not only changes the element content of metal, but also causes the variation in morphologies. The number of laser pulses contained in a single burst (N) has a direct effect on the surface morphologies. When N increases from 5 to 25, the surface morphologies are shown in Figure 2a1–c1. The LIPSS is uniformly distributed on the surface of NiTi, accompanied by a period shallow pit structure. This LIPSS–pit composite structure becomes more pronounced with the increase in N. The above-mentioned SEM image was further investigated through the 2D Fourier transformation, the corresponding results of which are shown in Figure 2a2–c2. In the intensity map of the Fourier transformation, the spatial frequency components corresponding to the nano-gratings become increasingly evident when the number of pulses increase from 5 to 15. As the number of pulses reaches 25, the ripples still seem evident on the 2D Fourier map but show more blurs in the Y direction compared with 15 pulses because the crater structures are also more apparent with the increase in pulse numbers. The results of the 2D Fourier map are in good agreement with the surface morphologies. The single pit was further measured by AFM. The results are shown in Figure 2a3–c3,a4–c4, which indicate that the formed pit was about 4 µm in diameter and 300 nm in depth when N = 5. When N was increased from 5 to 15, the diameter of pit remained unchanged and the depth gradually increased to 700 nm. When N was further increased from 15 to 25, the diameter of the pit remained consistent at about 4 µm, but the depth increased from 700 nm to 900 nm. These results show that the LIPSS width and the pit diameter of the metallic surface remained unchanged, but the depth of the pit gradually increased as N was increased. It is speculated that when N is low, the LIPSS effect dominates the formation of micro-nano structures on metallic surfaces [31]. At the same time, due to the uneven energy distribution of Bessel beams (the energy density of the middle core and its first surrounding ring is high, and the energy density of the higher orders of rings is low), the center of the ripples produces a deepening pit with the increase in N. 

As N is further increased from 50 to 250, the LIPSS–pit composite structure gradually evolved into a deep porous structure under the action of MHz burst Bessel beam (Figure 3a1–c1). The ripple structure around the micropores weakened with the increase in N from 50 to 250. At the same time, a circular deep groove structure appeared around the pores. In the process of increasing N, the fade of the ripples is clearly reflected on the Fourier transformation maps of the SEM images. When N = 50, the spatial frequency components corresponding to ripples were still evident (Figure 3a2), but faded quickly when N reached 100 (Figure 3b2). When N = 250, the ripples almost disappeared on the 2D Fourier transformation map (Figure 3c2). The cross-sectional morphologies of NiTi alloy were measured by focused ion beam (FIB). Figure 3a3–c3 shows that NiTi alloy formed a cone-shaped deep porous structure under the action of a single MHz burst Bessel beam. As N increased from 50 to 150 and then to 250, FIB results show that the depth of the cone-shaped pores increased from 4.1 µm to 8.2 µm and, finally, up to 10.5 µm. All of the superficial opening diameters of the cone-shaped holes were about 3 µm, and none of them changed significantly with the increase in N. The inner diameters of the cone-shaped holes were 1 µm, 1.4 µm, and 1.3 µm at N of 50, 100, and 250, respectively, but in the process of increasing N, the hole appeared to bend obviously. We speculate that this was caused by inhomogeneity within the materials. The destruction of the material inside the metal was a process of continuous accumulation with the incident pulse. The metal after laser pulse ablation became a hot fluid, and the hot fluid flowed along the softened damage area, actively flowing to the severely damaged part. Finally, the molten metal was recast inside. The unevenness of the surface and the inhomogeneous nature of the material caused the initial non-uniform ablation structure to accumulate with the increase in N due to the path-dependent effect, and, finally, form a curved deep hole structure inside the metal. At the same time, the reaction force exerted on the NiTi alloy by the plasma jet process caused the molten material to be extruded to form a splash and buildup of material, a portion of which was recast within the hole. On the inner wall of the deep hole, the accumulation of the recast still could be observed.

In order to explore the formation process of micropores on metal, the energy distribution inside the NiTi alloy of the blocked Bessel beam was fitted. The diffraction simulation results of the only-core propagation (all the surrounding rings were blocked in the incident plane) showed that the blocked Bessel beam core presented a diameter of 1.5 µm and only 1.9 µm in half maximum for the light intensity distribution (Figure 3d1,e). The light field acquired by the interference between the core and the first ring of the Bessel beam was also simulated (Figure 3d2,e). It was found that the intensity distribution of Bessel beam was still a cone shape with a diameter of 1.5 µm; however, the depth was increased to 5.5 µm for half of the maximum light intensity. The simulation results are in good agreement with the experimental results, which indicate that the interference between the core and the first ring of the Bessel beam were mainly responsible for the formation of pores. To sum up, it can be speculated that when N is small and the laser energy accumulation is weak, the LIPSS effect dominates the initial ablation process, thus producing LIPSS–pit composite structure. The LIPSS–pit composite structure provides free spaces for the downward propagation of the subsequent core and the first ring of the Bessel beam. With the increase in N, the ablation of the Bessel beam interference field dominates the process, the LIPSS gradually disappears, and the deep porous structure continues to extend downward, and the Bessel beam interference appears around the deep hole ablated ring structure. 

For exploring the universality of femtosecond MHz burst Bessel laser for the rapid preparation of wavelength-scale micropores on other metals, the same method was implemented on Al and SS, and similar micropores were also obtained. During the increase in N from 5 to 25, the SEM images (Figure 4a1–c1,e1–g1) showed that the LIPSS was evenly distributed on the surface of Al and SS. The LIPSS gradually evolved into shallow pit structures. The laser confocal microscopy results (Figure 4a2–c2,e2–g2) showed that the depth of the shallow pit on Al gradually deepened from 500 nm to 2 µm, and the depth of the shallow pit on SS also increased from 500 nm to 2 µm. As N was increased to 250, Al and SS surfaces formed a uniform porous structure. The porous dimeter of Al was 2 µm, and the porous dimeter of SS was 1.3 µm which was smaller than Al. Through cross-sectional observation of the metal porous structure (Figure 4d1–d2,h1–h2), it was found that the inner wall of the pores also presented a rough state. Further porous depth measurement revealed that the porous depth of Al was 10.4 µm. The porous depth of SS was 9.3 µm which was slightly shallower than Al. The porous depth and dimeter of Al were larger than SS, presumably, because the SS has a higher damage threshold than Al. The formation process of the porous structure of Al and SS was the same as that of NiTi alloy, both of which were LIPSS formed on the surface at low N and energy. LIPSS provided free space for the subsequent Bessel beam to propagate downward. With the increase in energy, the intense plasma ejection inside the metal formed a porous structure on the surface. This theory is expected to provide guidance to fabricate porous structures with higher aspect ratios on metal.

Porous metal can be applied in the preparation of anti-adhesion surfaces through lubricant injection. As shown in Figure 5a, the inside of porous metal stored lubricant and a lubricating layer formed on the upper surface of the metal. The lubricating layer inhibits the direct contact between metal and repelled liquids, thus endowing the anti-adhesion property of the metal. The dense porous structure of metal enhances the storage of lubricant. The wavelength-scale micropores improve the effect of the capillary force of micropores to the lubricant, thus improving the stability of the anti-adhesion property by inhibiting the lossless quality of the lubricant. The anti-adhesion property of the lubricant-infused porous metal was confirmed with common liquids, including milk, cola, coffee, and tomato juice. These droplets are ellipsoidal on metal and do not spread out completely, indicating the lubricant-infused porous metal has a certain of lyophobicity (Figure 5b). The dynamic sliding behavior of droplets on the surface of lubricant-infused porous metal also shows similar results. As shown in Figure 5c, various liquids quickly slide off the porous metal without leaving a trace, thus displaying the excellent anti-adhesion properties of modified metal. In a word, rapid fabrication of wavelength-scale micropores on metal is of great significance for the development of industry, medicine, and other fields.

## 4. Conclusions

The preparation of micropores on metallic surfaces attracts much attention due to the important applications in aerospace, biomedical, and industrial fields. At present, most diameters of micropores prepared on metallic surfaces are more than 20 µm, so that there are still great difficulties in the preparation of smaller wavelength-scale micropores on metallic surfaces. In this paper, a new method for rapid fabrication of wavelength-scale micropores on metallic surface by spatiotemporally shaped femtosecond beams is proposed. In the spatial domain, the Gaussian light generated by fiber laser is converted into Bessel light by an axicon. In the time domain, the pulse sequence is selected by external trigger to achieve the femtosecond MHz burst output. The number of pulses in a single burst can be adjusted by the high-level action in the square wave, and the frequency of the output pulse train can be regulated by the frequency of the square wave. Finally, high-density micropores with a dimeter of 1.3 µm and a depth of up to 10.5 µm were formed on metallic surfaces under the action of a femtosecond MHz burst Bessel beam. The movement of the translation stage can complete the preparation of 2000 micropores in 1 s, demonstrating the high efficiency of this method. Theoretical analysis and experimental results proved that the wavelength-scale micropores on metal are the result of the combined action of the LIPSS effect and Bessel beam interference. At the beginning of the laser action, the LIPSS–pit composite structure was produced under the LIPSS effect. The LIPSS–pit composite structure provides free space for the downward propagation of the subsequent Bessel beam interference. As the laser pulses continue to accumulate, the ablation of the Bessel beam interference field dominates the process, the LIPSS disappears, and the deep porous structure gradually extends downward to form a high-density wavelength-scale porous structure on metallic surfaces. The same phenomenon and results have also been verified on Al and SS, confirming that the method of preparing wavelength-scale micropores on metallic surfaces by femtosecond MHz burst Bessel beam ablation has wide universality. Prepared porous metal can store lubricant and use capillary force to cause the lubricant to adhere tightly, thus endowing the lubricant-infused porous metal with anti-adhesion properties. In the future, it is necessary to pay more attention to the quality of micropores, solve the problem of hole bending, and achieve controllable processing of size and morphology through light field regulation. The formation process of micropores on metallic surfaces provides a new physical insight into the rapid preparation of microporous structures with special dimensions on various metals, and promotes the application of porous metals in aerospace, medical, industrial, and other fields. 

## Figures and Tables

**Figure 1 nanomaterials-12-04378-f001:**
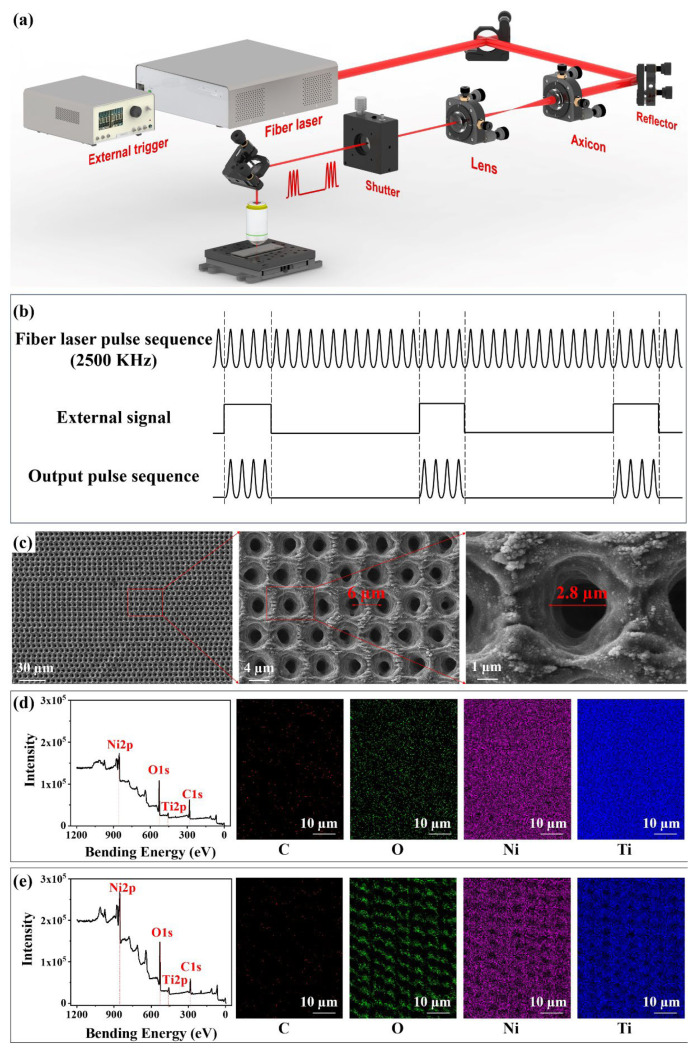
Schematic diagram and morphological characteristics of microporous processing on a metallic surface. (**a**) Optical path diagram of wavelength-scale micropores on metal. (**b**) Schematic diagram of the principle of controlling the output pulse train by external trigger. (**c**) Surface morphologies of microporous NiTi alloy. (**d**,**e**) Elements distribution and intensity of NiTi alloy (**d**) before and (**e**) after laser ablation.

**Figure 2 nanomaterials-12-04378-f002:**
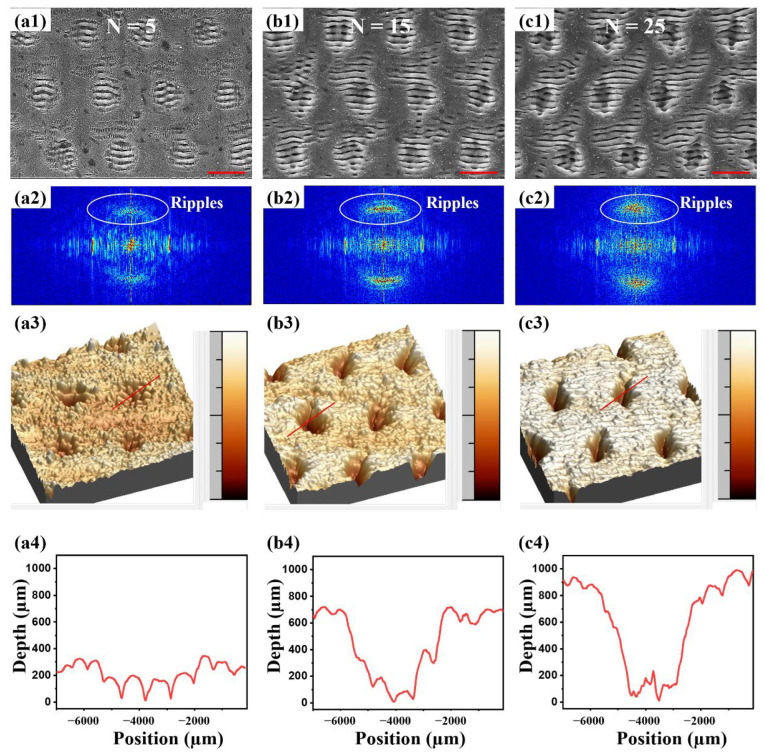
Surface morphologies of NiTi alloy under the action of burst Bessel beam with low N and Fourier transformation of SEM LIPSS–pit structure. (**a1**–**c1**) Surface morphologies of NiTi alloy under the action of burst Bessel beam with (**a1**) N = 5, (**b1**) N = 15, and (**c1**) N = 25. (**a2**–**c2**) 2D Fourier transformation of the SEM LIPSS–pit structure in (**a1**–**c1**). (**a3**–**c3**) 3D AFM of SEM LIPSS–pit structure in (**a1**–**c1**). (**a4**–**c4**) 2D AFM of single LIPSS–pit structure in (**a1**–**c1**). (Scale bar = 5 µm).

**Figure 3 nanomaterials-12-04378-f003:**
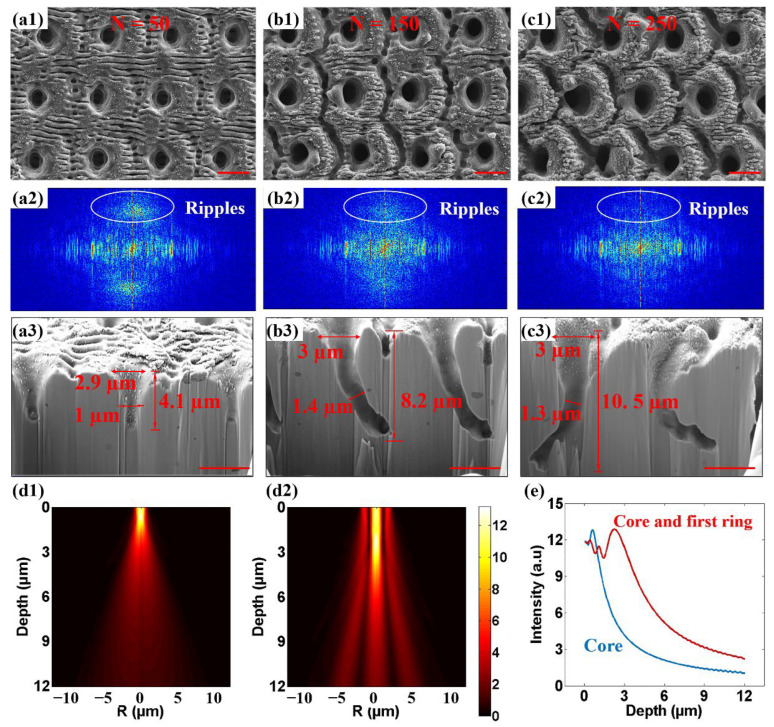
Surface morphologies of NiTi alloy under the action of burst Bessel beam with high N and Fourier transformation of SEM LIPSS–pit structure. (**a1**–**c1**) Surface morphologies of NiTi alloy under the action of burst Bessel beam with (**a1**) N = 50, (**b1**) N = 150, and (**c1**) N = 250. (**a2**–**c2**) 2D Fourier transformation of the SEM porous structure in (**a1**–**c1**). (**a3**–**c3**) Cross-sectional SEM of porous structure in (**a1**–**c1**). (**d1**–**d2**) Blocked Bessel beam propagation (**d1**) core only and (**d2**) the core and first ring of the Bessel beam. (**e**) Variation curve of Bessel beam intensity with depth of different orders. (Scale bar = 4 µm).

**Figure 4 nanomaterials-12-04378-f004:**
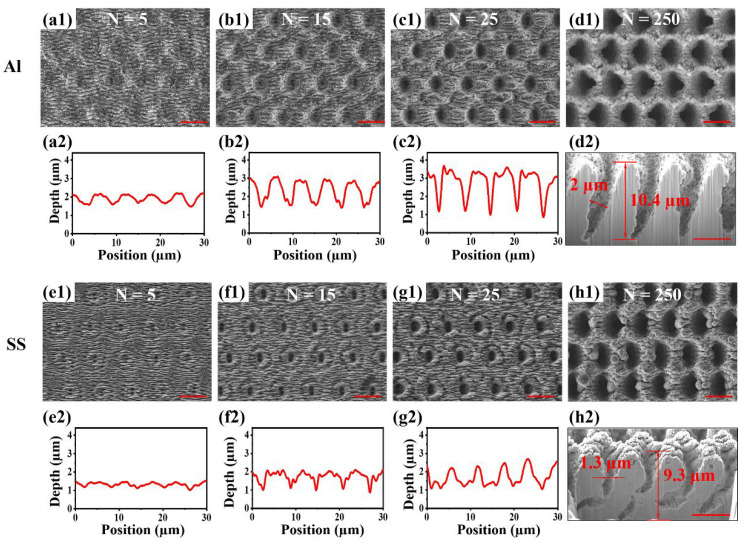
Surface morphologies of Al and SS after laser ablation. (**a1**–**d1**) Surface morphologies of Al under the action of burst Bessel beam with (**a1**) N = 5, (**b1**) N = 15, (**c1**) N = 25, and (**d1**) N = 250. (**a2**–**c2**) Laser confocal microscopy of LIPSS–pit structure in (**a1**–**c1**). (**d2**) The cross-section morphology of SEM in (**d1**). (**e1**–**h1**) Surface morphologies of SS under the action of burst Bessel beam with (**e1**) N = 5, (**f1**) N = 15, (**g1**) N = 25, and (**h1**) N = 250. (**e2**–**g2**) Laser confocal microscopy of LIPSS–pit structure in (**e1**–**g1**). (**h2**) The cross-section morphology of SEM in (**h1**). (Scale bar = 5 µm).

**Figure 5 nanomaterials-12-04378-f005:**
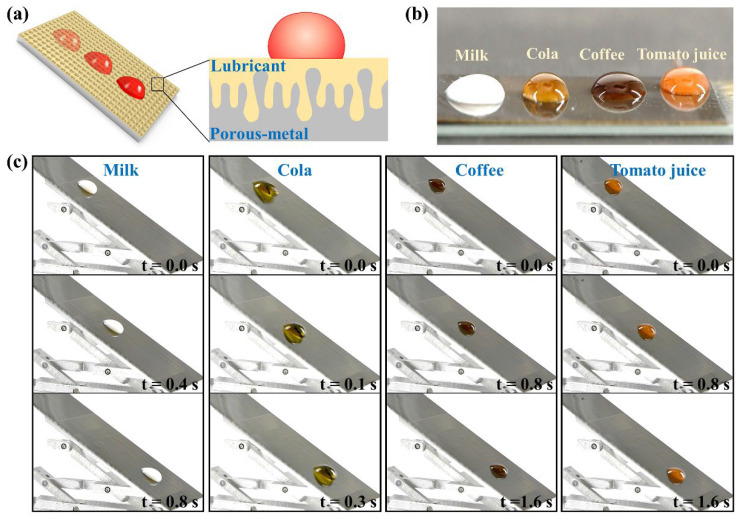
Anti-adhesion properties of lubricant-infused porous metal. (**a**) Anti-adhesion principle of lubricant-infused porous metal. (**b**) The static resistance of lubricant-infused porous metal to different liquids. (**c**) The dynamic resistance of lubricant-infused porous metal to different liquids.

## Data Availability

Not applicable.

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
