# Peer review of "Rapid Fabrication of Wavelength-Scale Micropores on Metal by Femtosecond MHz Burst Bessel Beam Ablation"

_nanomaterials, 2022, doi:10.3390/nano12244378_

Round 1

Reviewer 1 Report

"Rapid fabrication of wavelength scale micropores on metal by femtosecond MHz burst Bessel beam"

The Authors present and discuss some results concerning the formations of wavelength scale porous structures, as the result of the co-action of the laser induced periodic surface structure effect and Bessel beam interference. The manuscript is well written, yet several aspects need revision.

* Title: not meaningful, incomplete
* Keywords: meaningful, yet insufficient (provide more)
* Abstract: not meaningful, please insist more on the novelty and importance of your work and provide more relevant data; avoid redundant phrases; use present tense (recommended)

1.    Introduction

* this section is rather short, missing important information; the Authors need to insist more on the novelty and importance of their approach with respect to literature and their previous work (some important publications are missing): further explain on your approach, and also consider providing more / further references to it.

* in the last paragraph of the section, a brief presentation of the work herein needs to be provided, with sufficient and relevant details, insisting on the novelty and importance of your approach; please rephrase this section accordingly.

2.    Materials and Methods

* as a general remark, this section is too short (incomplete) and the experimental methods need to be further discussed; however, there's no need to present well-known techniques (references will suffice, please provide more where available); avoid redundant text; provide further information on the methods, working principles, compounds used, etc

3.    Results and discussion (please rename as such)

* as a general overview / remark to this section: the Authors need to further discuss their results in a more correlated manner; provide more references to sustain your results (where available).
* please further present and discuss in text all figures and tables, provide references to sustain your results (where available).
* a final (last) paragraph of section 3 must be included, to provide the reader with a brief conclusion of your work / manuscript (and insist more on the novelty of your approach).

4.    Conclusion

* the Authors need to emphasize more on the novelty and importance of this approach, providing sufficient data relevant to this study; remember that this method of using Bessel beams and LIPPS is neither "new", nor out of the ordinary (for details, please further refer to literature).

To conclude, the manuscript should be considered for publication only after careful revision.

Author Response

Reviewer 1

The Authors present and discuss some results concerning the formations of wavelength scale porous structures, as the result of the co-action of the laser induced periodic surface structure effect and Bessel beam interference. The manuscript is well written, yet several aspects need revision.

(1) * Title: not meaningful, incomplete
* Keywords: meaningful, yet insufficient (provide more)
* Abstract: not meaningful, please insist more on the novelty and importance of your work and provide more relevant data; avoid redundant phrases; use present tense (recommended)
  Reply: Thanks for your comments.

  1. The title has been revised to “Rapid fabrication of wavelength scale micropores on metal by femtosecond MHz burst Bessel beam ablation”.
  2. The revised key words are “femtosecond laser; MHz burst Bessel beam; porous metal; wavelength scale micropores; laser induced periodic surface structure (LIPSS).”
  3. The revised abstract as follows:

“The preparation of the wavelength scale micropores on metallic surfaces are limited by the high opacity of metal. At present, most of micropores reported by literatures are more than 20 µm in diameter, which is not only large in size, but also inefficiency in processing so that it is difficult to meet the needs of some special field, such as aerospace, biotechnology and so on. In this paper, the rapid laser fabrications of the wavelength scale micropores on various metallic surfaces are realized through femtosecond MHz burst Bessel beam ablation. Taking advantage of long depth focal field of the Bessel beam, high-density micropores with a diameter of 1.3 µm and a depth of 10.5 µm are prepared on metal by MHz burst accumulation. In addition, the rapid fabrication of 2000 micropores can be achieved in 1 second. The guidelines and experimental results illustrate that the formations of the wavelength scale porous structures are the result of the co-action of the laser induced periodic surface structure (LIPSS) effect and Bessel beam interference. Porous-metal can be used to store lubricant and form a lubricating layer on metallic surface, thus endowing the metal resistance to various liquids adhesion. The microporous formation process on metal provides a new physical insight for the rapid preparation of wavelength-scale metallic micropores, and promotes the application of porous-metal in the fields of catalysis, gas adsorption, structural templates, and bio-transportation fields.”

(2)  Introduction

* this section is rather short, missing important information; the Authors need to insist more on the novelty and importance of their approach with respect to literature and their previous work (some important publications are missing): further explain on your approach, and also consider providing more / further references to it.

* in the last paragraph of the section, a brief presentation of the work herein needs to be provided, with sufficient and relevant details, insisting on the novelty and importance of your approach; please rephrase this section accordingly.

 Reply: Thanks for you advises. Some new literature and work have been added. The last paragraph of introduction also gives a brief description of this work. The revised introduction has been highlighted in red, and the part of newly added sentences as follows:

“Lee et al. used femtosecond multi-pulses to fabricate micropores with diameter of 7µm on titanium implants to enhance the quality of the peri-implant soft-tissue integration.[27] Every spot was created by 2 × 104 pulses, and a spherical focusing lens with a numerical aperture of 0.39 was optimal for the fabrication of pores.”

[27] Abdal-hay, A.; Staples, R.; Alhazaa, A.; Fournier, B.; Al-Gawati, M.; Lee, R.S.; Ivanovski, S. Fabrication of Micropores on Titanium Implants Using Femtosecond Laser Technology: Perpendicular Attachment of Connective Tissues as a Pilot Study. Optics & Laser Technology 2022, 148, 107624, doi:10.1016/j.optlastec.2021.107624.

“At present, ultrashort pulse laser exhibits unique advantages in the preparation of precision micropores,[29–31] and the most of the micropores diameter prepared by femtosecond laser on the metallic surface is more than 20 µm, and the low processing efficiency is difficult to achieve rapid processing of large area.”

[29] Hu, M.; Ge, L.; Zhang, J.; Chen, Y.; Chen, X. Hole-Drilling with High Depth-Diameter Ratio Using Multi-Pulse Femtosecond Laser. Chin. J. Laser 2016, 43, 0403006, doi:10.3788/CJL201643.0403006.

[30] Xie, Q.; Li, X.; Jiang, L.; Xia, B.; Yan, X.; Zhao, W.; Lu, Y. High-Aspect-Ratio, High-Quality Microdrilling by Electron Density Control Using a Femtosecond Laser Bessel Beam. Appl. Phys. A 2016, 122, 136, doi:10.1007/s00339-016-9613-x.

[31] Kamlage, G.; Chichkov, B.N.; Ostendorf, A.; Toenshoff, H.K. Deep Drilling of Metals by Femtosecond Laser Pulses.; Phipps, C.R., Ed.; Taos, NM, September 9 2002; p. 394.

(3) Materials and Methods

* as a general remark, this section is too short (incomplete) and the experimental methods need to be further discussed; however, there's no need to present well-known techniques (references will suffice, please provide more where available); avoid redundant text; provide further information on the methods, working principles, compounds used, etc

Reply: Thank you for your suggestion, a more detailed supplement was added in the part of materials and methods. The following sentences and figure 1b are added in the revised manuscript page 2.

 “The processing diagram of porous metal is shown in figure 1a. A femtosecond fiber laser (FemtoYSL-20, YSL-photonics) generate Gaussian beam with the center wavelength of 1030 nm and the pulse width of 270 fs. Output femtosecond pulses is controlled by the square wave signal generated by external trigger as shown in figure 1b. When the external trigger is at a high level of 5 V, the laser outputs. When the external trigger is at a low level of 0 V, the laser stops outputting. Therefore, the number of laser pulses contained in a single burst (N) can be regulated by adjusting the duty of the square wave, and the frequency of the output pulse train can be regulated by adjusting the square wave frequency. Subsequently, the Gaussian beam is converted into Bessel beam through the axicon (Thorlabs, AX-250, α = 1°), and then converged into the objective lens (Nikon, ×20, NA=0.4) through lens.”

 (4) Results and discussion (please rename as such)

* as a general overview / remark to this section: the Authors need to further discuss their results in a more correlated manner; provide more references to sustain your results (where available).
* please further present and discuss in text all figures and tables, provide references to sustain your results (where available).
* a final (last) paragraph of section 3 must be included, to provide the reader with a brief conclusion of your work / manuscript (and insist more on the novelty of your approach).

Reply:Thanks for your advices. We have examined the manuscript in detail and revised it according your suggestions. The specific revises have been marked in red.

(5) Conclusion

* the Authors need to emphasize more on the novelty and importance of this approach, providing sufficient data relevant to this study; remember that this method of using Bessel beams and LIPPS is neither "new", nor out of the ordinary (for details, please further refer to literature).
To conclude, the manuscript should be considered for publication only after careful revision.

Reply: Although there are a lot of researches on the fabrication of micropores by femtosecond Bessel ablation, the research objects are mostly silicon wafers or transparent materials, and the diameter of the pores reported in literature is more than 20 µm, but there are few researches focused on the non-transparent metals. Herein, we focused on the preparation of wavelength scale micropores on metal by femtosecond MHz burst Bessel beam ablation. In addition, the methodd of preparing micropores on metal by femtosecond MHz burst Bessel beam ablation is universal. Limited by the speed of the processing stage, 2000 micropores can be obtained in one second, showing high processing efficiency. According to your suggestions, the advised conclusion as follows:

“The preparation of micropores on metallic surface attracts much attention due to its important applications in aerospace, biomedical and industrial fields. At present, the most of the micropores diameter prepared on the metallic surface is more than 20 µm, so that there are still great difficulties in the preparation of smaller wavelength-scale micropores on metallic surfaces. In this paper, a new method for rapid fabrication of wavelength scale micropores on metallic surface by spatiotemporally shaped femtosecond beams is proposed. In the spatial domain, the Gaussian light generated by fiber laser is converted into Bessel light by an axicon. In the time domain, the pulse sequence is selected by external trigger to realize the femtosecond MHz burst output. The number of pulses in a single burst can be adjusted by the duty of high level in the square wave, and the frequency of the output pulse train can be regulated by the frequency of square wave. Finally, high-density micropores with a dimeter of 1.3 µm and a depth of up to 10.5 µm formed on metallic surface under the action of femtosecond MHz burst Bessel beam. The movement of translation stage can complete the preparation of 2000 micropores in 1 second, showing the high efficiency of this method. Theoretical analysis and experimental results proved that the wavelength scale micropores on metal is the result of the combined action of LIPSS effect and Bessel beam interference. At the beginning of the laser action, the LIPSS-pit composite structure produced under the LIPSS effect. The LIPSS-pit composite structure provides free space for the downward propagation of the subsequent Bessel beam interference. With the laser pulses continue to accumulate, the ablation of Bessel beam interference field dominates the process, the LIPSS disappears, and the deep-porous structure gradually extends downward to form high-density wavelength scale porous structure on metallic surfaces. The same phenomenon and results have also been verified on Al and SS, confirming that the method of preparing wavelength-scale micropores on metallic surfaces by femtosecond MHz burst Bessel beam ablation has wide universality. Prepared porous-metal can store lubricant and use the capillary force to make the lubricant adhere tightly, thus endowing the lubricant infused porous-metal with anti-adhesion properties. In the future, it is necessary to pay more attention to the quality of micropores, solve the problem of hole bending, and realize the controllable processing of size and morphology through light field regulation. The formation process of micropores on metallic surfaces provides a new physical insight for the rapid preparation of microporous structure with special dimensions on various metal, and promotes the application of porous-metals in aerospace, medical, industrial and other fields.”

Special thanks for your helpful comments.

Reviewer 2 Report

The paper describes the production of micropores by burst of femtosecond pulses with a Bessel spatial distribution. The paper also describes the use of this method to produce anti-adhesion surfaces through lubricant injection. The paper is interesting and of general interest for readers of Nanomaterials, so I recommend its publication with minor corrections:

·     i) The writing language should be improved.

·    ii) From the SEM images of Figures 3b3, 4d2 and 4h2, it seems that the holes created by the laser beam are bended in the same direction. This is in contradiction with the authors hypothesis that this bending is due to inhomogeneities within the materials.

Author Response

Reviewer 2

The paper describes the production of micropores by burst of femtosecond pulses with a Bessel spatial distribution. The paper also describes the use of this method to produce anti-adhesion surfaces through lubricant injection. The paper is interesting and of general interest for readers of Nanomaterials, so I recommend its publication with minor corrections:

(1) The writing language should be improved.

Reply: Thank you for your advice. The writing of this manuscript was carefully improved. We have marked on the revised manuscript in red. We hope the improvement can meet the request.

(2) From the SEM images of Figures 3b3, 4d2 and 4h2, it seems that the holes created by the laser beam are bended in the same direction. This is in contradiction with the authors hypothesis that this bending is due to inhomogeneities within the materials.

Reply: Thank you for your question. We also have the same confusion about hole bending. At first, we guessed that it was related to the movement direction of the sample, but the experiment found that the adjacent holes showed different bending directions. In addition, the cross-section width of micropores is only 22 µm, which is so small that it is difficult to identify the regular of the hole bending direction. Finally, we assumed that it is caused by the non-uniformity of the sample, but there is no experimental evidence for this speculation, so we plan to find out the real reason in the following work, and we included the problem of bending holes in the future work plan. The following sentences were added in the conclusion:

“In the future, it is necessary to pay more attention to the quality of micropores, solve the problem of hole bending, and realize the controllable processing of size and morphology through light field regulation.”

Special thanks to your helpful comments.

Reviewer 3 Report

Dear authors,

thank you for you high quality work.

Please, make little correctionse:

1. Line 169-170. Why diapasone is 50-250?

2. Materials and Methods. Please, give more details about setting of the equipment for your work.

3. Add in the end of Conclusion announcer for your next research steps of this topic.

Best regards,

Reviewer

Author Response

Reviewer 3

thank you for you high quality work. Please, make little corrections:

(1) Line 169-170. Why diapason is 50-250?

Reply: Thanks for your comment. When N=50, the deep hole structure starts to become obvious. With the increase of N, the micropores gradually deepening and the LIPSS phenomenon gradually weakening. When N=250, the hole depth is maximum and the surface appears to be circular. If N continues to increase, the increase of the number of single pulse train will cause the width of the holes become larger and show an ellipse, which will cause the overlapping of adjacent holes. Therefore, the number of pulses is selected in the range of 50-250.

(2) Materials and Methods. Please, give more details about setting of the equipment for your work.

Reply:Thank you for your suggestion, a more detailed supplement was added in the part of materials and methods. The following sentences and figure 1b are added in the revised manuscript:

 “The processing diagram of porous metal is shown in figure 1a. A femtosecond fiber laser (FemtoYSL-20, YSL-photonics) generate Gaussian beam with the center wavelength of 1030 nm and the pulse width of 270 fs. Output femtosecond pulses is controlled by the square wave signal generated by external trigger as shown in figure 1b. When the external trigger is at a high level of 5 V, the laser outputs. When the external trigger is at a low level of 0 V, the laser stops outputting. Therefore, the number of laser pulses contained in a single burst (N) can be regulated by adjusting the duty of the square wave, and the frequency of the output pulse train can be regulated by adjusting the square wave frequency. Subsequently, the Gaussian beam is converted into Bessel beam through the axicon (Thorlabs, AX-250, α = 1°), and then converged into the objective lens (Nikon, ×20, NA=0.4) through lens.”

(3) Add in the end of Conclusion announcer for your next research steps of this topic.

Reply: Thank you for your advice, the next research plan has added in the conclusion, the details as follows:

“In the future, it is necessary to pay more attention to the quality of micropores, solve the problem of hole bending, and realize the controllable processing of size and morphology through light field regulation.”

Special thanks to you for your helpful comments.

Round 2

Reviewer 1 Report

"Rapid fabrication of wavelength scale micropores on metal by femtosecond MHz burst Bessel beam" original title

"Rapid fabrication of wavelength scale micropores on metal by femtosecond MHz burst Bessel beam ablation" - revision 1

The Authors have correctly addressed most of the issues raised during the peer-review procedure. The manuscript is now suitable for publication.